# Geosite Assessment in the Beigua UNESCO Global Geopark (Liguria, Italy): A Case Study in Linking Geoheritage with Education, Tourism, and Community Involvement

**Pietro Marescotti** [1,*], **Giulia Castello** [2], **Antonino Briguglio** [1], **Maria Cristina Caprioglio** [2], **Laura Crispini** [1] and **Marco Firpo** [1]

1  Department for the Earth, Environment and Life Sciences (DiSTAV), University of Genova, 16132 Genova, Italy
2  Parco del Beigua UNESCO Global Geopark, 17019 Varazze, Italy
*  Correspondence: pietro.marescotti@unige.it

**Abstract:** The inventory and the assessment of geosites plays a very important role in highlighting scientific, geotouristic, and geoeducational potential, as well as the ability to identify any criticalities and vulnerabilities of the geological heritage of a territory. Within a geopark, these assessment activities are also crucial for developing land management strategies and policies that not only meet the need to protect geological and natural heritage, but also to promote sustainable economic development of the area and local communities. The Beigua UNESCO Global Geopark (Liguria, Italy) includes fifty-four sites known for their significant geological values. In this work, we have combined a study aimed at the qualitative and quantitative evaluation of 10 of the 54 sites with the results of an analysis of the educational, touristic, and land management activities that have been developed on these sites from 2011 to 2021. The quantitative assessment of the ten selected sites reveals their high scientific value and considerable touristic and/or educational potential. Thus, they represent not only scientific geological heritage to be preserved but also a significant tourism resource for the geopark territory. This is confirmed by the great success of geotouristic and geoeducational initiatives developed in the park over the last ten years, and by the growing involvement of the local communities, institutions, entrepreneurial activities, as well as environmental, sports, and cultural associations. These results highlight some important aspects for the management of geological heritage and associated values within a geopark.

**Keywords:** corals; European Geopark Network; geoeducation; geotourism; ophiolites; serpentinites

## 1. Introduction

The Beigua Geopark is located in Liguria, in the north-western part of Italy (Figure 1). It is important for understanding the geological history of Italy and the Mediterranean area and is characterized by a remarkable geological heritage due to an extraordinary geodiversity with outstanding scientific, geoeducational, and geotouristic values [1]. In March 2005, the Beigua Geopark was awarded the status of European and Global Geopark under the aegis of the European Geopark Network (EGN) assisted by UNESCO. In November 2015, it was designated as a UNESCO Global Geopark (UGGp) and in October 2021, the designation was confirmed after the revalidation process. To date, fifty-four sites of geological interest have been recognized in the park territory. They include many geosites with significant geomorphological, petrological, mineralogical, paleontological, stratigraphic, structural, hydrogeological, and/or paleogeographical values (as reported in the UGGp and Beigua Geopark websites [2,3] and described in detail in Section 4.1 "Inventory and qualitative assessment of selected geosites from Beigua UGGp" of this manuscript). To date, twelve of fifty-four sites have been officially recognized as geosites (i.e., sites of geological

importance) in the National Inventory by the Italian Istituto Superiore per la Protezione e la Ricerca Ambientale (ISPRA) [4] for their high national and international scientific values.

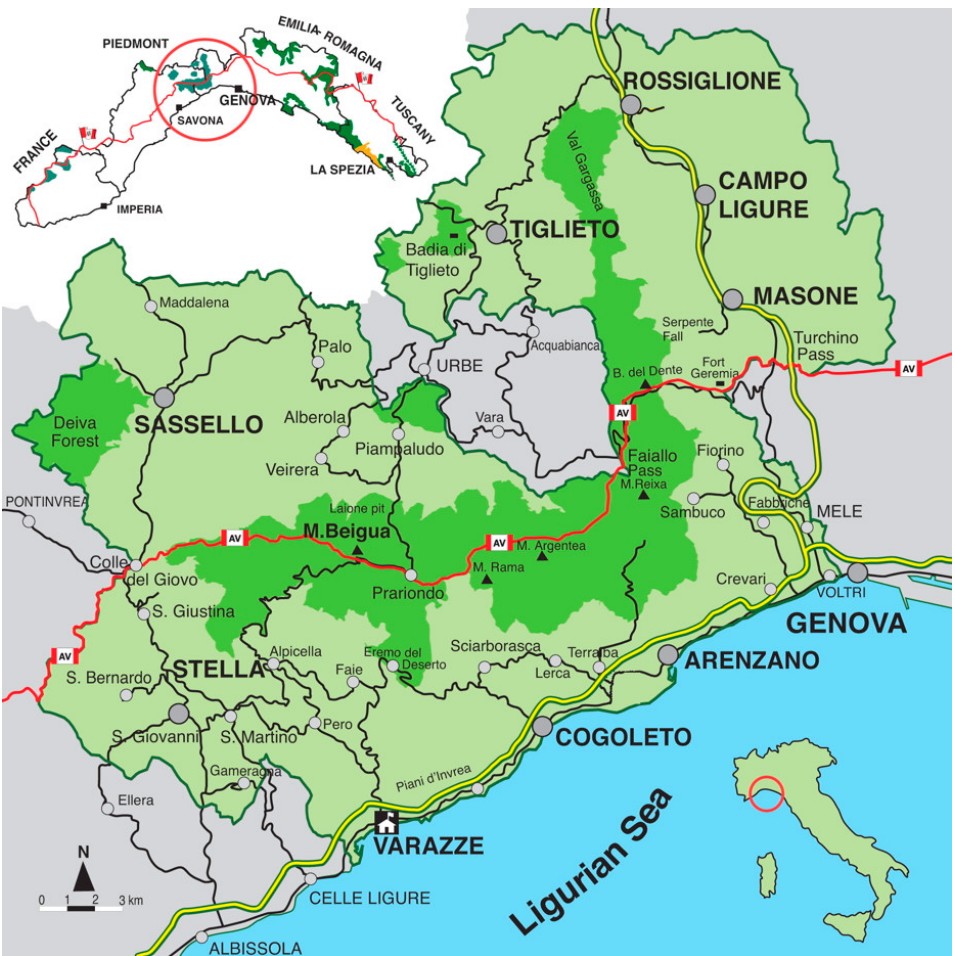

**Figure 1.** Geographic location of Beigua Geopark territory. Dark green: Beigua Regional Nature Park; Light green: Beigua UGGp. The red line is the "Alta Via dei Monti Liguri" which roughly coincides with the Tyrrhenian–Padanian watershed. In the upper-left inset, the green areas are the Regional Parks of Liguria and the yellow area is the Cinque Terre National Park. Adapted with permission from Ref. [1].

According to the official definition of the EGN [5,6] a "*geopark is a territory which combines the protection and promotion of geological heritage with sustainable local development*". Therefore, it is important for geopark authorities to develop a set of land management strategies and policies that not only meet the need to protect geological and natural heritage, but also to promote sustainable economic development of the area and local communities.

In order to achieve this goal, the assessment and monitoring of geosites should be continuously developed within the territory of a geopark to highlight both scientific, geo-touristic, and geoeducational potential [7–10], as well as any criticalities and vulnerabilities.

In the last two decades several methods and criteria for the qualitative and quantitative assessment of geosites and, in particular, of geomorphosites have been proposed [11–32]. The analysis of the different proposed methods shows that the inventory and preliminary qualitative assessment of geosites and sites of geoheritage significance should take into consideration: (*i*) the key geology of the area; (*ii*) the scale of reference of geoheritage features (i.e., from megascale to very fine scale), their level of significance (i.e., international, national, regional, local), and the linkages to inter-related ensembles of geological features [26], (*iii*) all the geographic and geological information of the site to evaluate the

scientific value, the educational and touristic potential, as well as the additional values (such as aesthetic, ecological, and cultural values) which can allow highlighting the relationship between geology and natural and human environment, e.g., [7,14,30]; specifically for geotourism, several methods of site selection have been developed, since not all sites with geoheritage values are suitable for tourism, e.g., [12,24,30]. In fact, several specific values are required for these sites, including aesthetics, emotional value, authenticity, uniqueness, visual value, accessibility, safety, and support services, e.g., [7,33,34]; (iv) the vulnerability of the site and any actions to be taken to protect significant sites by natural or anthropogenic degradation, e.g., [35].

Once an inventory and the qualitative assessment have been conducted, most of the information and other specific attributes can be quantified by attempting to score the different values and, more generally, the site under consideration according to objective, measurable, and reproducible criteria e.g., [7,12–17,24].

In this work, we have combined a study aimed at the qualitative and quantitative evaluation of ten selected geosites from the Beigua UGGp with the results of an analysis of the educational, touristic, and land management activities that have been developed on these sites from 2011 to the present day in collaboration with institutions, entrepreneurial activities, as well as environmental, sports, and cultural associations.

The results of this comparison are also important in assessing the potential societal impact of geological heritage, because they provide a numerical feedback based on the success of initiatives undertaken within the geopark over the past ten years.

## 2. Geographical and Geological Background

The Beigua UGGp is located in the Ligurian Alps, covers an area of approximately 39,000 hectares, and has a boundary that includes the whole territory of the Beigua Regional Nature Park, involving ten municipalities (Arenzano, Campo Ligure, Cogoleto, Genova, Masone, Rossiglione, Sassello, Stella, Tiglieto, Varazze) and two provincial districts (Genova and Savona) (Figure 1). The geopark is easily reachable by State roads from the main highways. A good range of main and side roads is available inside the area. Moreover, the eastern and the southern side are served by the national railways and public transportation is available all over the geopark. In the geopark, there is a trail network of approximately 500 km of varying difficulty suitable for hikers and bikers.

The Beigua UGGp is managed by the Beigua Regional Nature Park Authority, that is, a public equivalent institution founded in 1996 (Regional Law n. 12/1995) and supervised by the Regional Administration of Liguria. The Beigua Park Authority has the responsibility for all necessary actions towards the benefit of the area in combination with the protection of the natural park and of the four sites of the European NATURA 2000 Network (Birds Directive and Habitats Directive). Moreover, it develops and promotes actions of study, research, promotion, exhibition, conservation, protection, and sustainable fruition of the park.

The park authority has two offices, one administrative in the Municipality of Sassello and one operational in the Municipality of Varazze. Moreover, thanks to the several Visitor Centers and Information Points located in the territory, the park provides touristic–naturalistic welcome services with the distribution of informative material and information on the regulations and on the use within the protected area. Among these are the Palazzo Gervino's Museum and Park House (Sassello), the Experience Centre and Information Point in Varazze, the Ornithological and Environmental Education Centre in Arenzano, the Villa Bagnara's Visitor Center in Masone, and the Information Points in Tiglieto and Cogoleto.

The territory of the park is crossed from west to east by the Tyrrhenian–Padanian watershed, with a marked asymmetry between the two slopes of the ridge: the Tyrrhenian slope, with higher acclivity, whereas the Padanian side has a lower one [36]. The Ligurian–Tyrrhenian watershed runs for approximately 25 km from the Giovo of Sassello hill (Savona) to the Turchino pass (Genova) (Figure 1) and divides the geopark area into a southern and a northern side. In this context, the Beigua massif represents a spectacular natural balcony

formed by mountains (most of them with elevations over 1000 m a.s.l.) overlooking the Ligurian Sea (Figure 2). The main relief in the area is represented by Mt. Beigua (1287 m), Mt. Reixa (1183 m a.s.l.), Mt. Rama (1150 m), Bric del Dente (1107 m a.s.l.), Mt. Sciguelo (1103 m a.s.l.), and Mt. Argentea (1082 m a.s.l.).

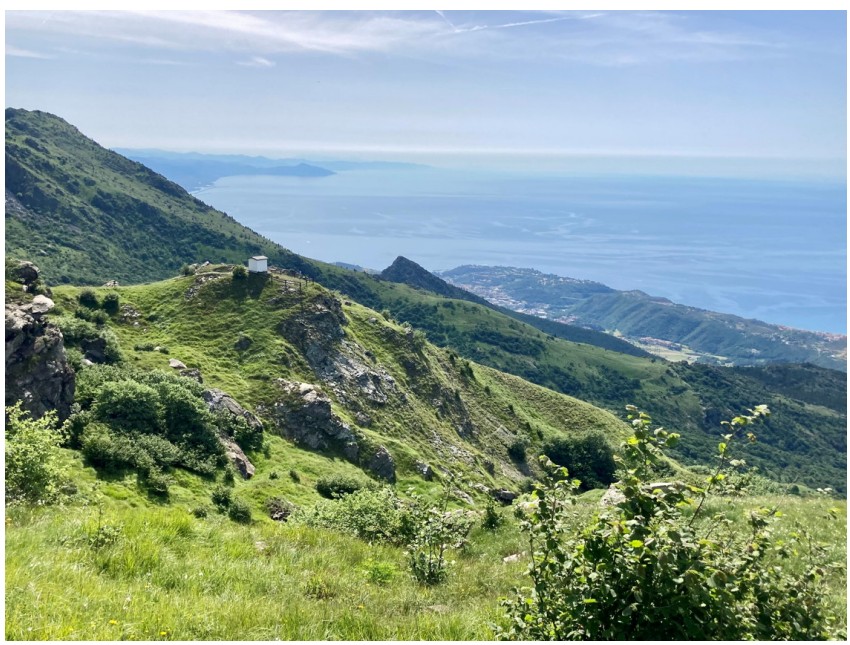

**Figure 2.** Panoramic view of the Tyrrhenian slopes from Pratorotondo (Mt Beigua).

The water courses on the Tyrrhenian slope are poorly hierarchical, short in length, with rather high average gradients, and are oriented transversely to the coastline. They have a torrential character and are fed exclusively by rainfall, with variable flow rates and flood events in autumn [36]. On the contrary, the hydrographic network of the Po Valley slope appears to be better-developed and hierarchical; the main streams and creeks (Erro, Orba, Gargassa streams) are mostly set on faults and tectonic lineaments, and are characterized by irregular path, deeply etched grooves, and entrenched meanders [36].

Beigua UGGp can be divided in three major geological–morphological domains: a southern coastal region facing the Ligurian Sea; a central sector dominated by the reliefs of the Tyrrhenian–Padanian watershed, and a north-western region that comprises former exhumed sectors of the orogenic belt partially subsided during Oligocene–Miocene to form sediment-filled basins.

As for geologic features, Beigua UGGp is located at the junction between the Ligurian Alps (southeastern termination of the Italian Western Alps) and the Northern Apennines and is characterized by outstanding and unique geodiversity resulting from the long and complex geologic history of its rocks. It is mostly composed of ophiolites, including part of their oceanic sedimentary cover, with minor occurrences of metamorphosed rocks of continental crust (gneiss and sedimentary carbonate successions); all the lithologies are capped by limited outcrops of clastic sedimentary rocks and Quaternary sediments (Figure 3).

The Beigua UGGp meta-ophiolite (i.e., Voltri Massif *Auct.*) is among the remnants of the Tethyan ophiolites in the Mediterranean area, and one of the main ophiolitic complexes of the Italian Alps–Apennine system. It contains fragments of oceanic crust and upper mantle of the Jurassic ocean basin (Ligurian Tethys; e.g., [37]) formed by seafloor spreading after the pre-Triassic rifting of the Europe–Adria continental domain (ca. 200–145 Ma; e.g., [38]). According to the paleogeographical and tectonic reconstructions, e.g., [39,40], and references therein], fragments of the Europa and Adria continents were piled up with slices of the oceanic rocks during the Cretaceous–Eocene convergent events of the Alpine Orogeny (spanning ca. 145–33 Ma), during which they acquired metamorphic overprint and struc-

tural deformations. The Voltri Massif ophiolite has been recrystallized at various pressures and temperatures during this long-lasting polyphase metamorphic history and preserves records either of the ocean-floor metamorphism (amphibolite to greenschists facies conditions) or of the HP-LT subduction–exhumation Alpine events (blueschist to eclogite facies peak conditions) [41–44]. Serpentinite in the Beigua UGGp area includes metagabbro bodies of various size (km- to m- scale), lenses of metabasalt, and minor metarodingite dikes [45]. The stack of these tectono-metamorphic units was then unconformably overlain by late to post-orogenic clastic sediments of the Upper Eocene to Miocene Tertiary Piedmont Basin (a syn-tectonic Neoalpine–Apennine basin, e.g., [46,47]).

Afterwards, from the Neogene to the Quaternary, the whole area was shaped by the geological events related to the opening of the Ligurian–Provençal back arc basin and the rotation of the Corsica–Sardinia block (mainly faulting and backthrusting [48]), and by the Plio-Quaternary uplifts and tilting of the Ligurian Alps at the seaward side, e.g., [49].

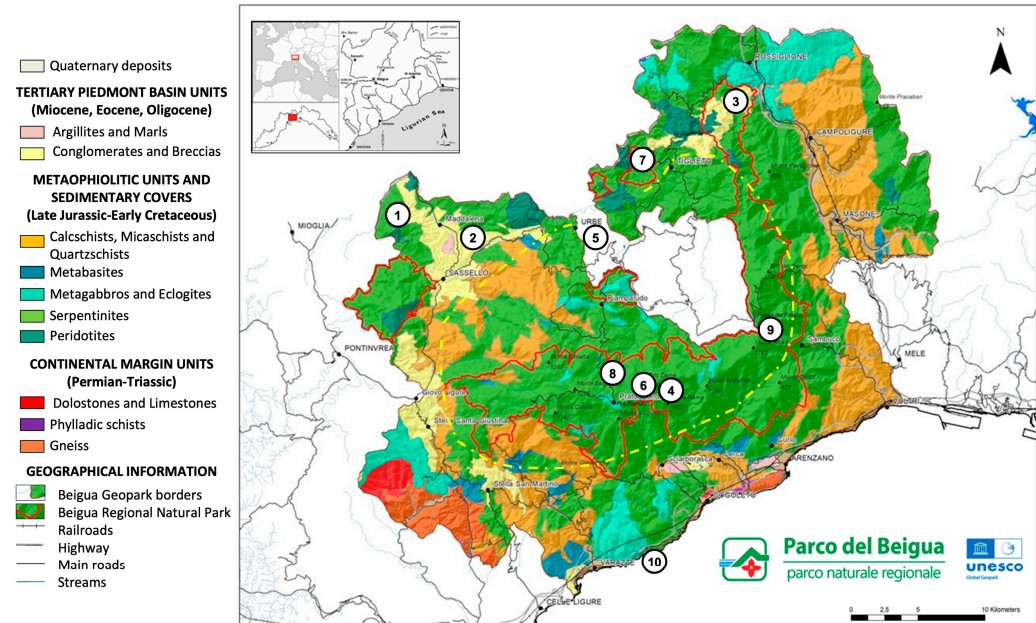

**Figure 3.** Simplified geological sketch of the Beigua Geopark area (Modified from "Geotouristic map of Beigua Geopark"); adapted with permission from Ref. [50]. The yellow dashed line indicates the area where eclogite and metarodingite lenses and dikes of various scale commonly occur within the meta-ophiolite. The numbers (1–10) indicate the location of the selected geosites described in the text.

## 3. Materials and Methods

Among the fifty-four sites of geological interest recognized in the Beigua UGGp, we selected five of the twelve geosites officially listed in the National Inventory [4] and five geosites chosen as representative of both different geological aspects and major educational and tourism activities carried out in the park over the past decade. For each selected site, we performed a qualitative and quantitative assessment based on the methods and criteria proposed by Brilha (2016) [7].

The evaluation procedures, both qualitative and quantitative, were based on direct and indirect methods consisting of in situ observations and on a review of existing literature, thematic cartography, and reports produced by geopark guides and operators from 2005 to date.

The inventory and the qualitative assessment of geosites included in the Italian inventory by ISPRA [4] comprise (Table 1): (*i*) geographical information (geographical coordinates, municipality, legal protection, accessibility) (*ii*) scientific geological information (geological description, geological features with scientific value, geological framework, scientific knowledge, limitations to scientific use, representativeness, integrity, rarity);

(*iii*) other potentials (didactic and interpretative potential, observation conditions, scenery);
(*iv*) main fragility and vulnerability.

**Table 1.** Inventory and qualitative assessment of selected geosites (G) from Beigua UGGp included in the Italian inventory by ISPRA [4] according to the criteria of Brilha (2016) [7].

| Geosite (Locality) | (G1) Peridotitic Spheroids (Lago dei Gulli) | (G2) Fossil Corals (Ponte Prina) | (G3) Val Gargassa Canyon (Gargassino) | (G4) Blockfields (Pian del Fretto) | (G5) Eclogites and Metarodingites (Vara) |
|---|---|---|---|---|---|
| Geographical Coordinates (WGS84) | 44°29′23.3″; 8°27′51.6″ | 44°29′53.9″; 8°29′53.41″ | 44°33′1″; 8°39′26″ | 44°25′52.6″; 8°36′17.7″ | 44°28′37.6″; 8°35′52.8″ |
| Municipality | Sassello (Savona) | Sassello (Savona) | Rossiglione (Genova) | Sassello (Savona) | Urbe (Savona) |
| Owner | Private | Private | Public | Private | Private |
| Legal Protection | International (92/43/EEC, 1992) | Regional (hydrogeological restrictions) | International (92/43/EEC, 1992) | International (92/43/EEC, 1992; EEC 79/409, 1979) | Regional (hydrogeological restrictions) |
| Accessibility | Very good (Provincial route) | Very good (Provincial route) | Good (hiking trail) | Good (hiking trail) | Very good (Provincial route) |
| Geological description | Metric lherzolitic spheroids hosted by serpentine-schist and cataclastic serpentinites | Patchy reefal body with rich coralline rhodophyte, mollusks, and foraminiferal diversity | Fluvial canyon developed on Late Eocene–Early Oligocene polygenic conglomerates | Hectometric deposit of boulder- or block-sized angular rocks mostly represented by serpentinites. | Hectometric eclogite lenses and scattered metarodingite lenses and dikes hosted in antigoritic serpentine-schists. |
| Geological features with scientific value | Petrological, structural, geomorphological | Paleontological, paleogeographical, stratigraphical, palaeoclimatological | Hydrogeological, petrological, geomorphological, stratigraphical, palaeoclimatological | Geomorphological, paleogeographical, palaeoclimatological | Petrological, mineralogical, structural |
| Geological framework | Tectono-metamorphic Alpine ophiolites | Late to post-orogenetic, sedimentary succession deposited in the Tertiary Piedmont Basin (late Oligocene) | Late to post-orogenetic sedimentary succession deposited in the Tertiary Piedmont Basin (late Oligocene) | Periglacial deposit originated from cryoclastic processes during the last Pleistocene glacial period | Tectono-metamorphic Alpine ophiolites |
| Scientific knowledge | High (international) | High (international) | High (international) | High (international) | High (international) |
| Limitations to scientific use | None | Coral colonies are sporadic, and sampling should be avoided or very limited | None | None | None |
| Representativeness | Excellent. Well represented lherzolite relics within tectono-metamorphic serpentinites | Well represented late Oligocene shallow water sequence with peculiar paleoenvironmental characteristics | Excellent. Well represented sedimentary lithologies with alpine ophiolitic clasts and fluvial erosional landform | Excellent. Very extensive deposit representative of periglacial processes during Pleistocene glaciation | Excellent. Extensive outcrops of eclogites and metarodingites with evident relationships with serpentinitic host rocks |
| Integrity | Well preserved outcrop | Moderately preserved outcrop (threat of erosion and partial vegetation overgrowth) | Well preserved outcrops and geomorphological and hydrogeological features | Well preserved blockfield deposit | Well preserved outcrops |
| Rarity | Very rare occurrence | Very rare, especially the direct colonization by coral colonies on serpentines | Rare (particularly the Molare polygenic conglomerates) | Moderately rare (several examples in the Beigua UGGp) | Moderately rare |
| Didactic potential | Secondary school and university | Primary and secondary school, university | Primary and secondary school, university | Primary and secondary school, university | University |
| Interpretative Potential | Good | Very good | Good | Very good | Moderate |
| Observation conditions | Very good (on site, panoramic) | Very good (on site) | Very good (on site, panoramic) | Very good (on site, panoramic) | Very good (on site) |
| Scenery | High | Medium | Very high | High | Moderate |
| Fragility and vulnerability | Mean (hydrogeological) | High (hydrogeological, anthropic) | Mean (hydrogeological) | Low | Low (although there is a potential economic interest for titanium exploitation) |

The inventory and the qualitative assessment of the other geosites include (Table 2): (*i*) geographical information (same as for geosites) (*ii*) geological information (geological description, geodiversity features with potential educational and/or geotouristic uses, links

with ecological and cultural assets, limitations to didactic or touristic use); (*iii*) didactic and touristic potentials (didactic and interpretative potential, observation conditions, scenery and touristic potential); (*iv*) main fragility and vulnerability.

**Table 2.** Inventory and qualitative assessment of selected geosites (GD) from Beigua UGGp identified as of geological significance, but that are not currently on the ISPRA inventory [4], according to the criteria of Brilha (2016) [7].

| Geosite (Locality) | (GD6) Serpentinites (Pratorotondo) | (GD7) Orba Meander (Tiglieto) | (GD8) Laione Peat bog (Piampaludo) | (GD9) Garnet Crystals (Faiallo) | (GD10) Serpentinites and Metagabbros (Lungomare Europa) |
|---|---|---|---|---|---|
| Geographical Coordinates (WGS84) | 44°25′37.9″; 8°35′38.1″ | 44°31′33″; 8°35′58″ | 44°26′45.5″; 8°34′44.1″ | Non-disclosable | 44°21′53.6″; 8°36′12.5″ |
| Municipality | Sassello (Savona) | Tiglieto (Savona) | Sassello (Savona) | Urbe (Savona) | Varazze (Savona) |
| Owner | Private | Private | Private | | Public |
| Legal Protection | Regional (Natural Park restrictions) | International (92/43/EEC, 1992) | Regional (Natural Park restrictions) | Regional (hydrogeological restrictions) | Regional (hydrogeological restrictions) |
| Accessibility | Very good (hiking trail) | Very good (Provincial route) | Very good (Provincial route) | Medium (trekking path) | Very good (seafront promenade; pedestrian and bicycle path) |
| Geological description | Extensive outcrops of Jurassic serpentinites with various textural and structural features | Fluvial morphology with panoramic view of entrenched meanders | The main wetland in the park. It is the result of a small lake basin in a senescent state. In the site is also present a blockstream deposit | Large variety of garnets, with diverse morphologies, size, and colors, occurring within metarodingites and serpentinites | Extensive outcrops of metagabbros lenses within serpentinites |
| Geodiversity with potential educational and/or geotouristic uses | Excellent views of metamorphosed and metasomatized mantle rocks, along a scenic trail with several outstanding panoramic viewpoints along the Thyrrenian–Padanian watershed | Excellent example of natural and human influence on fluvial morphological evolution | Excellent example of link between geodiversity and biodiversity within wetland and peat bog deposit | Excellent example of metamorphic and metasomatic mineral assemblages (garnets, vesuvianite, titanite, apatite, diopside, and chlorite) | Several examples, along a scenic trail, of a tectono-metamorphic ophiolitic sequence (e.g., the relationships between mantle rocks and oceanic intrusive bodies) |
| Links with ecological and cultural assets | Botanical species with high pythogeographical significance (serpentinophytes or serpentinicolous relicts, microthermal orophilous, hygrophilous and endemic species). Use of building rock materials and ornamental stones. | Tiglieto Cistercian Abbey (1120 a.D.) | Wetland with high biodiversity and ecological value | None | None |
| Limitations to didactic or touristic use | None | None | None | None | None |
| Didactic potential | Secondary school and university | Secondary school and university | Primary and secondary school, university | Primary and secondary school, university | Primary and secondary school, university |
| Interpretative Potential | Good | Very good | Very good | Good | Very good |
| Observation conditions | Very good (on site, panoramic) | Very good (panoramic) | Very good (on site) | Very good (on site) | Very good (on site, panoramic views even from boat excursions) |
| Scenery and touristic potential | Very high | High | Moderate | Moderate | Very high |
| Fragility and vulnerability | Low | Low | High (dependent on changing climatic conditions) | High (illicit mineral sampling) | Low |

The quantitative assessment of all selected geosites includes four sections (Table S1) [7]: (*i*) scientific value (SV; 7 criteria); (*ii*) potential educational use (PEU; 12 criteria); (*iii*) potential touristic use (PTU; 13 criteria); (*iv*) degradation risk (DR; 5 criteria). Every criterion of the PEU, PTU and DR sections is ranked with a score from 0 to 4, whereas the criteria of SV are rated with the scores 0, 1, 2, 4 (i.e., not including score 3) in order to

better distinguish geosites ranked with 4 points. The scores were applied according to the indication reported in Brilha (2016) [7]. The final score for every section is a weighted sum calculated with the weights reported in Table S2. Finally, in addition to qualitative and quantitative assessment of the selected sites, an analysis of official data on educational and tourism activities developed from 2011 to 2021 was undertaken.

## 4. Results

### 4.1. Inventory and Qualitative Assessment of Selected Geosites from Beigua UGGp

The geosites included in the Italian inventory by ISPRA [4] (G1–G5; Table 1, Figures 3 and 4) and the other five geosites (GD6–GD10; Table 2, Figures 3 and 4) selected for this study are representative of the main geological features of the Beigua UGGp. All sites have good accessibility and can be reached via provincial roads or hiking trails of low to medium difficulty.

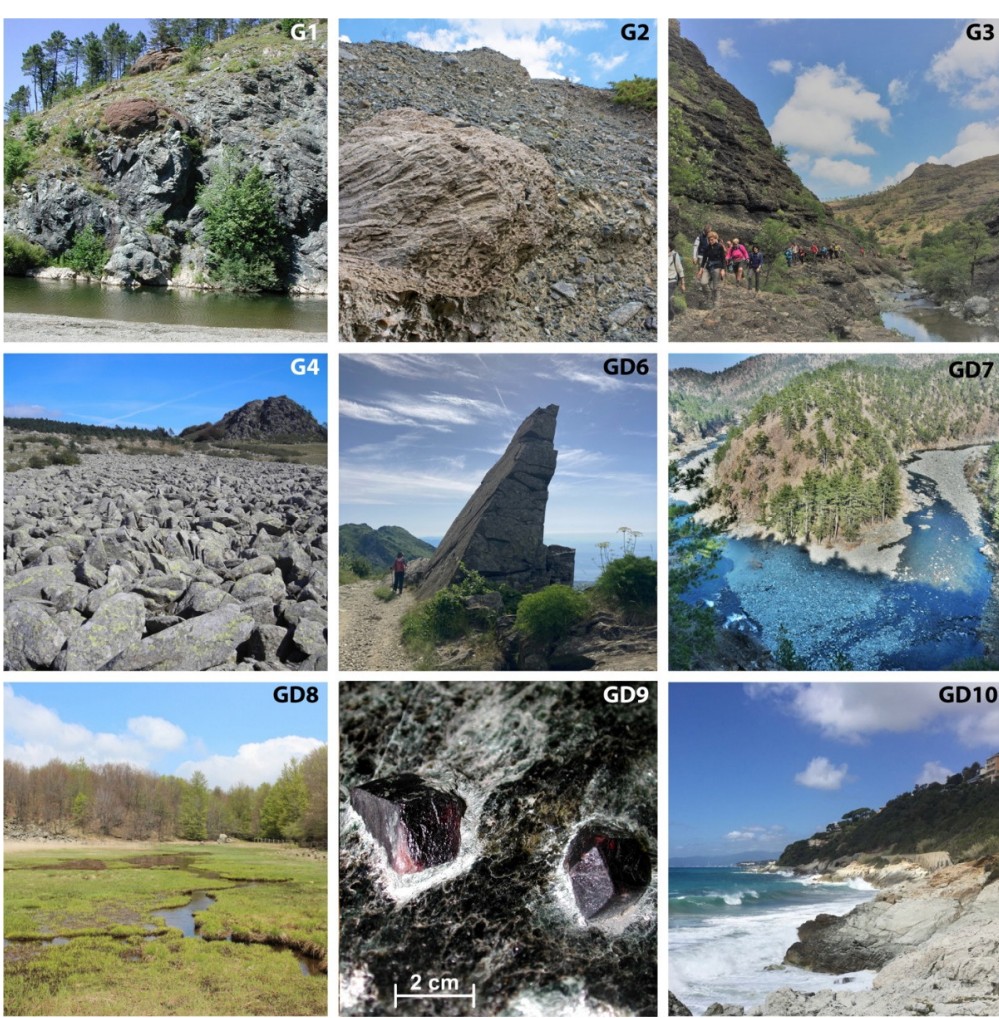

**Figure 4.** Selected views from the studied geosites. The geological features and other details of the sites are reported in Tables 1 and 2.

Five sites (G1, G5, GD6, GD9, GD10) are characterized by extensive and well-exposed outcrops of peridotites, serpentinites, metagabbros, eclogites, metabasites and metarodingites, each with significant scientific and/or educational value related to various mineralogical, petrological, structural, and palaeogeographical aspects. The scientific value of these sites is quoted by the relevant international and national literature dealing with: (*i*) the processes that drove extension and rifting of the continental Europe–Adria lithosphere, e.g., [37]; (*ii*) the geology of the Western Alps–Northern Apennine junction area,

e.g., [40] and more in general the overall architecture of the Alpine orogen, e.g., [39]; (*iii*) the subduction and exhumation processes during alpine orogenesis and the related metamorphic prograde and retrograde P–T paths of metamorphism, e.g., [41,43–45,51–54]. Furthermore, there are several scientific implications concerning the development of peculiar soils on ultramafic rocks, e.g., [55], as well as several links to biodiversity and ecological aspects.

Four sites (G3, G4, GD7, GD8) are particularly important from a geomorphological, hydrogeological, paleogeographical, and palaeoclimatological point of view; in particular: (*i*) the huge extension of blockfield and blockstream deposits in sites G4 and GD8 representative of periglacial processes during Pleistocene glaciation, e.g., [56]; (*ii*) the excellent examples of fluvial landforms, such as the entrenched meanders of GD7 and the impressive canyons developed on Late Eocene–Early Oligocene polygenic conglomerates of G3 [57]. Several links to cultural assets are present in the area of these sites that relate to (*i*) the medieval glassworks in Val Gargassa; [58]), (*ii*) the filigree craftsmanship, developed in Campo Ligure since the end of the 19th century, (*iii*) the important trade routes developed between the coast and the Po Valley [1], (*iv*) the ironworks in the Stura and Orba Valleys, and (*v*) the Tiglieto Abbey, the first Cistercian abbey founded outside France in 1120.

The last site (G2) is characterized by a late Oligocene sedimentary transgressive sequence that spans from branching and massive coral colonies encrusting directly on the Jurassic serpentinites, to foraminiferal and coralline-rich deposits, to riverine sediments smothering the reefal bioconstruction. This site has been studied extensively from both a paleontological and stratigraphic perspective, e.g., ([59] and references therein) because of its (*i*) primary importance for the reconstruction of the Oligocene geological and paleoenvironmental evolution of central Liguria and southern Piedmont, (*ii*) abundant and well-preserved high-diversity fossil content, (*iii*) very rare direct colonization by corals on serpentinites, and (*iv*) unique evidence of resilience of reefal systems to the globally recognized late Oligocene warming event (LOW) [60,61].

With the exception of G5, all the sites studied in this work have good or very good interpretative potential (Tables 1 and 2) and are particularly suitable for educational activities with students from schools of all levels and universities. Among them, G1, G3, G4, GD6, GD7, and GD10 are located in areas with a high to very high touristic potential.

The most important fragility and vulnerability are linked to the hydrogeological risk that is becoming increasingly high also due to major climate variations, which in the study area are leading to long periods of drought with very violent, prolonged, and localized storm events. Furthermore, sites G2 and GD9 also have a high vulnerability related to possible illegal sampling.

### 4.2. Quantitative Assessment of Selected Geosites from Beigua UGGp

The results of the quantitative assessment of the geosites selected for this work provided an overview of their scientific, educational, and touristic potential, as well as highlighted the risk of their degradation (Tables S1 and S2; Figure 5).

As expected, the highest weighted scores for scientific value (SV) were obtained for the five geosites G1–G5 (ranging from 280 to 320; Table S2, Figure 5) because they represent the best example in the study area to illustrate geological elements or processes related to the geological framework under consideration (i.e., high representativeness). In addition, these sites have been and still are the subject of numerous scientific research published in relevant international and national journals (see for example the references given in Section 4.1). The main discriminating factor in the final score among these geosites is the rarity of occurrence, not only in the park area but in some cases also in an international or national context (e.g., the geosite G2 for the very rare direct colonization by coral colonies on serpentines and for the peculiar paleoenvironmental characteristics of late Oligocene shallow water sequence).

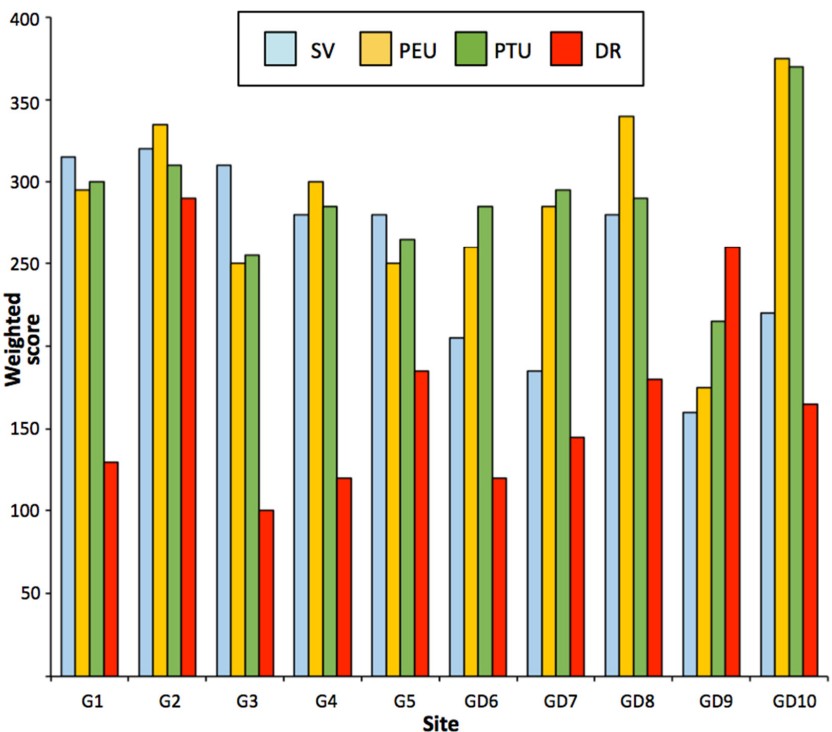

**Figure 5.** Weighted scores for the selected geosites. SV: scientific value; PEU: potential educational use; PTU: potential touristic use; DR: degradation risk. The description of the sites is reported in Tables 1 and 2. The tables with the results of the quantitative assessment and the weighted scores are reported in supplementary material (Tables S1 and S2, respectively).

Geosites that are not on the ISPRA inventory (Table 2) have significantly lower scores for SV (160–220), except for Site GD8 ("Laione peat bog"; 280). This site (GD8) should be considered for inclusion among the park's geotourism geosites as it represents the main wetland in the park resulting from a small lake basin in a senescent state. Moreover, within the site is also present an hectometric blockstream deposit with scientific relevance.

Weighted scores for the potential educational use (PEU) are fairly homogeneous and high for most of the sites considered (G1, G3–G7), ranging from 250 to 300 (Table S2, Figure 5). The highest weighted scores were obtained for G2 (335), GD8 (340) and, in particular, GD10 (375), due to the very high didactic potential and geological diversity values. In fact, this site, in addition to having extensive outcrops of metagabbro lenses within serpentinites, combines several elements of geological significance suitable for teaching at all levels of education, such as the outstanding marine terraces testifying to the Quaternary coastal evolution between Arenzano and Varazze [62,63]. The lowest score was obtained for site GD9 ("Garnet crystal from Faiallo"; 175). This site is internationally well known for the diffuse occurrence of garnets (mainly grossular and hydrogrossular) with a wide variety of morphologies, dimensions, and colors e.g., [64–66], and for this reason, garnets were extensively collected for their beauty and value for several years. Still recently, illegal sampling has been carried out despite regional laws prohibiting their collection throughout the entire geopark territory. For these reasons, the site no longer has significant exposures of geologic features that can be readily used for educational purposes and, due to its high vulnerability, it should be removed from the fifty-four sites currently recognized by the Beigua UGGp. The museum and expositions are the better location to highlight the scientific and didactic values of these geological elements, and several outstanding specimens from this site are exhibited at the Beigua UGGp Visitor Center of Palazzo Gervino (Sassello), at the Natural History Museum G. Doria (Genova), and at the Department for the Earth, Environment and Life Sciences of the University of Genova.

Results for the potential touristic use (PTU) evidenced high to very high scores for most of the selected sites (255–370; Table S2, Figure 5), with the highest values obtained by site GD10, not only for its interpretative potential, but also due to accessibility, proximity to recreational areas, scenery, and economic level of the municipalities included in the coastal sector between Arenzano and Varazze (this is, in fact, the only site studied occurring in the coastal sector of the geopark). Once again, the GD9 site has the lower score (215) for the same reasons described above (high vulnerability, low interpretative potential, non-optimal observation conditions).

According to the classification of Brilha (2016) [7], the degradation risk (DR) for eight of the ten sites studied (G1, G3–GD8, GD10) is low, the resulting scores being comprised between 130 and 185 (Table S2; Figure 5). In contrast, the DR of sites G2 and GD9 is rated as moderate (290 and 260, respectively); their scores are mainly due to the possibility of deterioration of the main geological elements due to the vulnerability to anthropic actions (e.g., uncontrolled sampling of fossil and minerals). In particular, the G2 site is located less than 500 m from a paved road in an area with legal protection but no access control.

*4.3. Analysis of the Activities Carried out from 2011 to 2021 Promoted by Beigua UGGp: Geoeducational Activities, Geotourism, and Community Involvement*

The Beigua UGGp offers a wide range of geological and environmental education activities for students and teachers in schools of all levels (Figure 6). The catalogue is updated every year with new proposals and includes short-term and long-term projects.

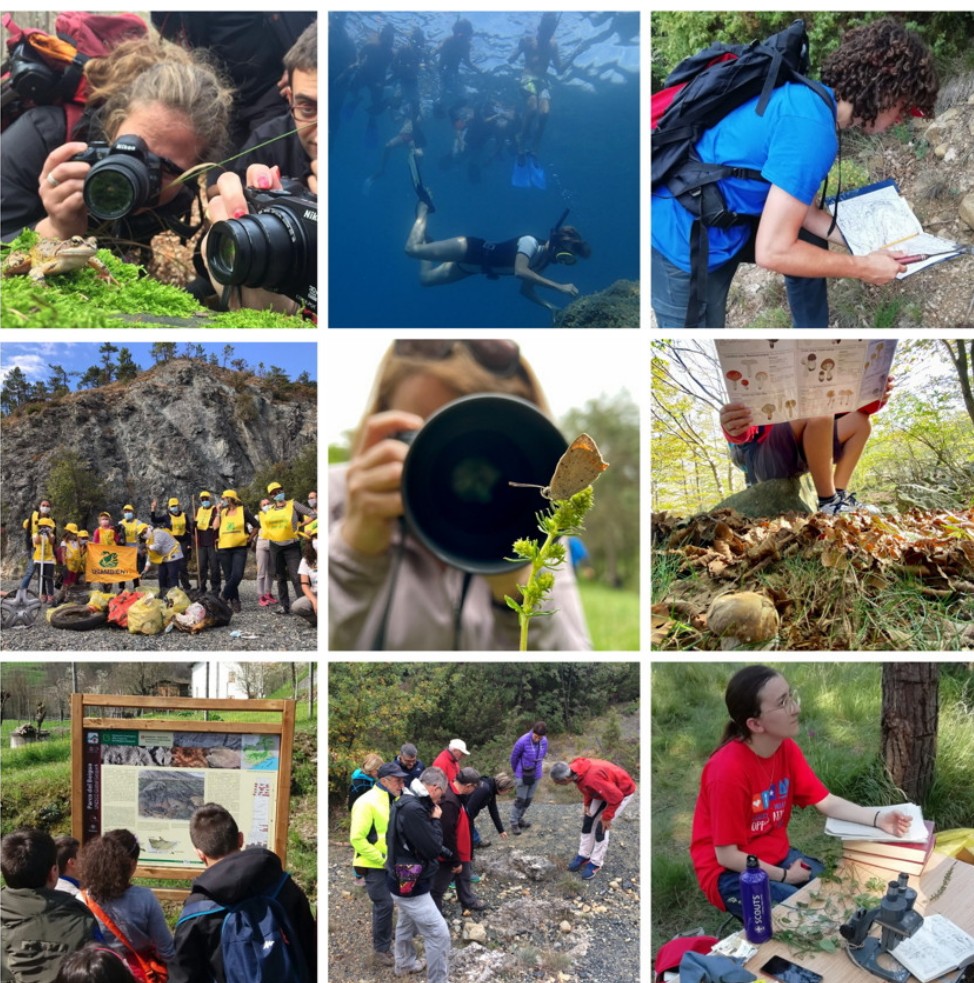

**Figure 6.** Views of the educational and touristic activities developed at the Beigua UGGp.

The results of the analysis of official data on educational activities developed from 2011 to 2021 in the sites selected for this study are shown in Figure 7. A total of 22,390 students participated in the educational activities promoted by Beigua UGGp in this timeframe, with a mean of 2035 students per year (range 607–3552). The distribution of participants in the different years considered is fairly homogeneous apart from the peak of over 3500 participants in 2011. This peak is related to a didactic project entitled "Citizens of the Park" and financed by the Liguria region, which involved, for one year, the primary and secondary students of six schools of the Sassello Comprehensive Institute. Conversely, the significant reduction in visitors in the years 2020 and 2021, can be entirely attributed to the considerable limitations imposed by the COVID-19 restrictions during the spring, as well as part of the summer and autumn periods. This was confirmed by the upward trend in visitors in 2021, following the analysis of the data for the first half of 2022, thus suggesting that, barring any new restrictions, participant numbers should soon return to the established levels of the pre-pandemic years. Ninety per cent of the students who participated in the activities of the decade analyzed are from primary and secondary schools and only ten per cent are from high secondary schools. The numbers quoted do not include university students, although field teaching activities are developed every year for students of Earth Sciences and Natural and Environmental Sciences at the University of Genova (approximately 100–200 students per year).

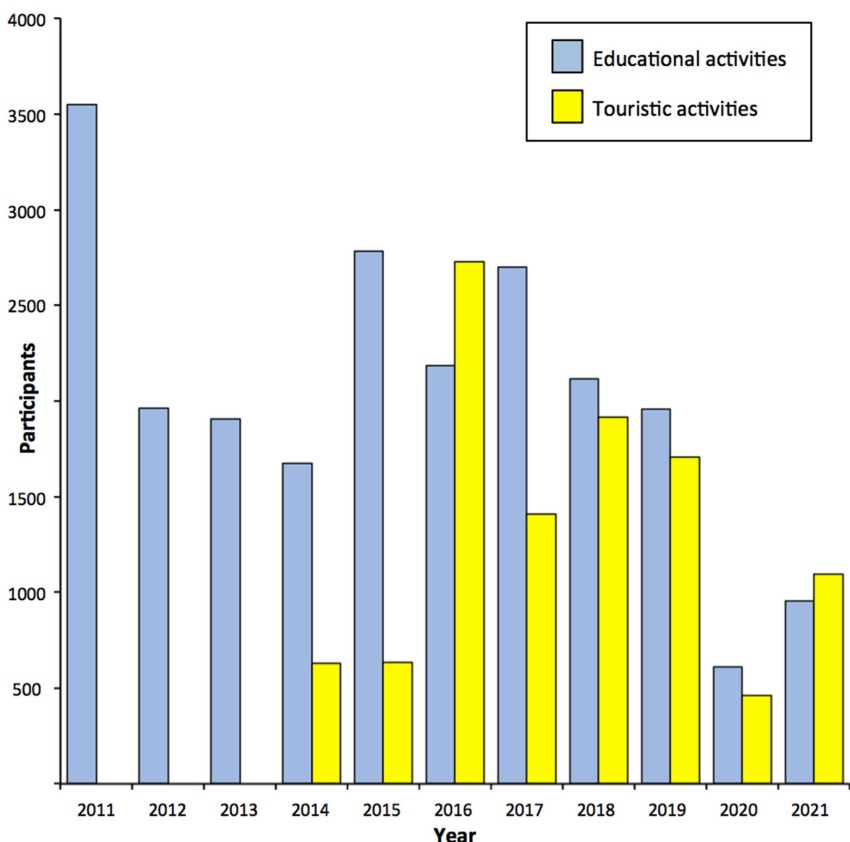

**Figure 7.** Number of participants in official activities organized by Beigua UGGp from 2011 to 2021. Data of participants in tourism activities from 2011 to 2013 are not available.

Regarding touristic activities, the geopark guides propose a calendar of weekly excursions aimed at promoting the area and enhancing its geological and natural features through activities and thematic visits focusing on various geological, environmental, and cultural aspects (Figure 6).

Data on people who participated in organized geotourism activities are only available from 2014 (Figure 7). The number of participants rose rapidly from over 600 in 2014 and

2015 (627 and 631, respectively) to 2731 in 2016 and then to between 1400 and 1900 in the three-year period 2017–2019. Once again, the considerations expressed above for the 2020 and 2021 pandemic period apply.

To these data must be added all hikers and bikers who use the network of trails that include the analyzed geosites. By way of example, two eco-counters along the paths of sites G3 and GD6 reported an average annual inflow of 49,106 passers-by (range over ten years 35,511–77,470). Extending these data to the entire set of sites studied (i.e., to those not covered by eco-counters), it can be assumed that the total number of tourists visiting the study area is at least five to ten times higher.

Finally, the geopark promotes the participation of local communities in the political and program decisions of the Beigua UGGp and continuously develops initiatives to increase the involvement of entrepreneurial activities, as well as environmental, sports, and cultural associations.

About these latest initiatives, in 2003, the park authority launched a specific study to identify the organoleptic properties of the different varieties of honeys produced in the geopark territory. To date, the brand "Honey of Beigua Park", consisting of a numbered guarantee label, is awarded to 6 producers that met a set of criteria including: (*i*) territoriality, (*ii*) production chain, (*iii*) hive characteristic and location, (*iv*) production, extraction, and processing methods, (*v*) packaging, for which pasteurization is forbidden.

In 2015, the "Tasty by Nature" brand was launched to acknowledge the close link between the geopark territory and the local fresh or processed food products. The brand is a promotional tool that neither identifies nor overlaps with the quality marks established by EU, national, or regional regulations; it is awarded to local agri-food enterprises that, with their activities, keep alive the traditions of the territory and carry out activities in support of the environmental protection actions promoted by the geopark authority. For the consumer, the brand represents both a guarantee of high-quality local products and a way of perceiving the intrinsic value of a specific product in terms of territorial uniqueness. To date, thirty-eight local enterprises have obtained the label for 130 different products representative of biodiversity and local history and traditions.

In 2018, a new brand, "Friendly by Nature", was launched for the accommodation facilities in the geopark municipalities. The eighteen accommodation facilities that have been awarded the label to date guarantee the quality of hospitality and the promotion of the geopark by providing materials and information on the geological and natural environment, hiking and cycling trails, and touristic activities focusing on geological, natural, and cultural heritage.

## 5. Discussion and Conclusions

Geoparks are key areas for communicating geological knowledge and gradually bringing people closer to a knowledge of the territory, its potential, and its fragilities, particularly in rural areas, in the light of the new concepts of green economy and sustainable development. These objectives can be pursued primarily through geotourism and geoeducational activities to be developed and planned after a careful assessment based on data that include the scientific value, the educational, and touristic potential, as well as the risk of degradation of geosites and geology of the territory under consideration. [7,11–35].

The results of this study highlight some important aspects for the management of geological heritage and associated values within a geopark. The quantitative assessment of the ten selected sites revealed a significantly higher scientific value of geosites included in the Italian inventory by ISPRA [4] than the other geosites, except for site GD8 (Figures 5 and 8). This is consistent with their definitions [4,7]; in fact, geosites included in the Italian inventory must have, as a primary condition, a recognized scientific value in terms of representativeness, integrity, rarity, and scientific knowledge.

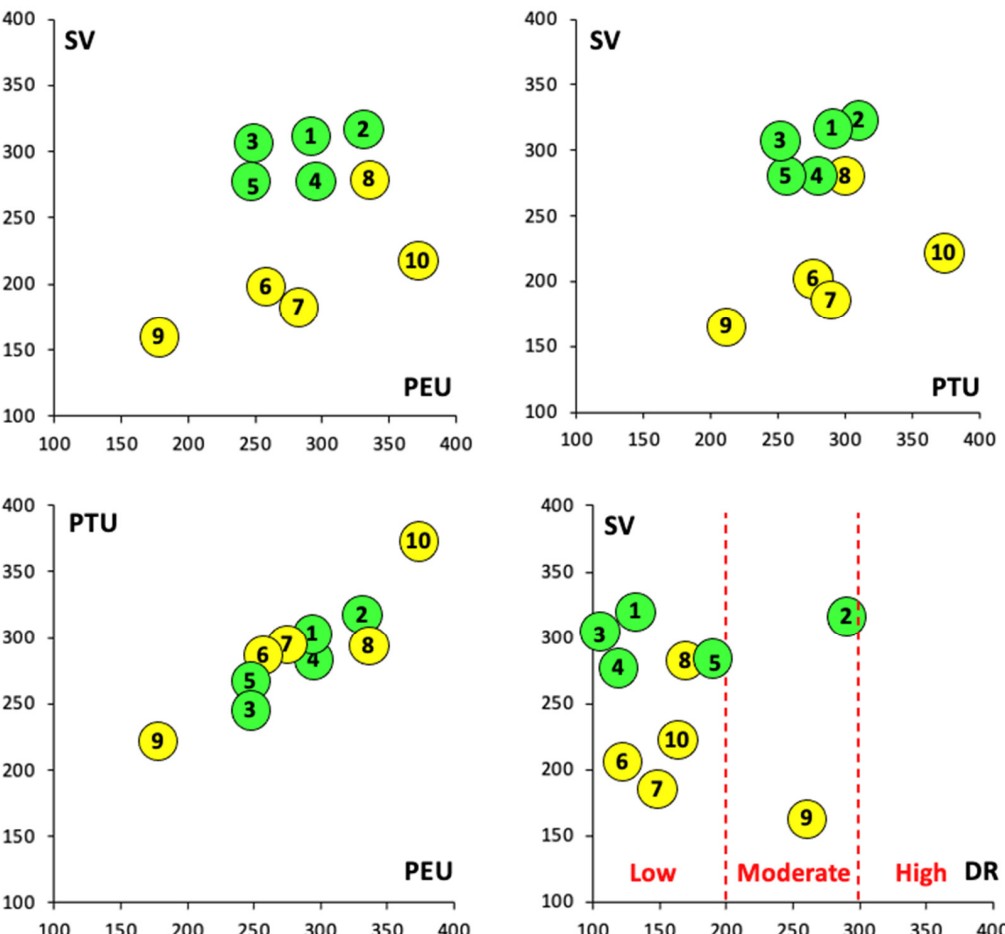

**Figure 8.** Comparison between the scientific value (SV), potential educational value (PEU), potential touristic value (PTU), and degradation risk (DR) of the geosites included in the Italian inventory by ISPRA [4] (green) and the other geosites (yellow) selected for this study. In the DR graph, the risk classes proposed by [7] are also reported.

Moreover, all the studied geosites have considerable touristic and/or educational potential, as well as low vulnerability and degradation risk (with the exception of the paleontological geosite G2). For this reason, they represent not only a scientific geological heritage to be preserved, but also a significant resource for the geopark. These results are also confirmed by the analysis of the geoeducational and geotouristic activities carried out in the Beigua UGGp over the last decade (Figure 7, Table 3).

**Table 3.** Percentages of educational and geotouristic activities carried out at the sites considered for this study from 2011 to 2021.

|  | G1 | G2 | G3 | G4 | G5 | GD6 | GD7 | GD8 | GD9 | GD10 |
|---|---|---|---|---|---|---|---|---|---|---|
| Geoeducational activities (%) | 3.13 | 25 | 4.38 | 3.75 | 0 | 2.50 | 1.88 | 9.38 | 0 | 50 |
| Geotouristic activities (%) | 1.82 | 5.45 | 9.09 | 20 | 0 | 20 | 9.09 | 16.36 | 7.27 | 10.91 |

All the sites considered were successfully used for the activities proposed by the geopark organization. The only exception is site G5, despite the considerable scientific value of the eclogite and metarodingite outcrops occurring in the area. This is mainly caused by the low didactic and interpretative potential for school students and lay people, as well as by its geographical location, which lacks scenically valuable elements, particularly in comparison to the other sites considered in this work. For sites with recognized scientific value, but with features not easily interpretable in situ, it is useful to provide appropriate

teaching material for school students and non-specialists. In particular for the eclogites and metarodingites of site G5, user-friendly panels and brochures explaining, from the macroscale to the very fine scale, the composition and the evolution of the rocks within the tectonic plate framework could help people understand their importance and unicity, as well as their peculiar physical properties. Moreover, a thematic geotrail (e.g., "the tectono-metamorphic evolution of an ophiolite complex") connecting this geosite to the other geosites in the area should be designed and implemented.

The high vulnerability and degradation risk of sites G2 (fossil corals from Ponte Prina, Table 1) and GD9 (garnet crystals from Faiallo, Table 2) are mainly related to geovandalism, i.e., to unauthorized collection of fossil and mineral specimens, e.g., [34] and references therein. Although regional laws prohibited their collection throughout the entire geopark territory, for these sites and, in particular, for the G2 geosite, management strategies should be developed; even sampling for scientific purposes should be carried out according to appropriate protocols that aim to preserve the integrity of the site before its geological relevance is irreparably compromised.

Another element that emerges from this analysis is the significant numerical difference between the educational and tourist activities developed at the GD10 site compared to all the other sites considered. This difference is mainly due to the geographical position of the site, which is the only one located in the coastal sector of the geopark along a pedestrian and bicycle path that joins three important tourist resorts of the Ligurian Riviera (i.e., Varazze, Cogoleto, and Arenzano). In this regard, it is important to emphasize that the territory of the Beigua UGGp and, more generally, the entire Liguria region, is characterized by a marked contrast in economic and demographic development between the coastal part and the inland rural areas. Along the Ligurian coast are several world-class tourist sites and seaside resorts that are visited by hundreds of thousands of national and international tourists throughout the year.

Several studies had shown that the proximity to other touristic attraction represents an important added value for the valorization of rural areas and can be an efficient driver driving force for geoheritage tourism, e.g., [29,67,68]. In order to create synergetic and innovative actions to attract visitors with different interests and to promote public knowledge about geology [69,70], it would be necessary to develop an effective geotourism management plan and a broader outreach strategy involving not only the geopark municipalities but also the regional authorities and business companies.

In addition to actions related to tourism development, the UNESCO Global Geoparks have a crucial role in promoting geosciences education through educational programs specifically addressed to the school community [71]. The success of the educational activities promoted by Beigua UGGp, with a mean of 2035 students per year (data from 2011 to 2021), emphasizes the growing interest of students and teachers in geological knowledge and, more generally, in the related issues of environmental sustainability and preservation of natural heritage. The only negative note is the low participation of high school students, which represent only 10% of the total. This aspect is at least partly related to the progressive reduction in the space dedicated to geology in Italian high school science programs and to the low percentage of teachers with a Master's degree in geological sciences among high school science teachers. Every year at the end of August, the Beigua UGGp publishes a catalogue and organizes a meeting with school operators and teachers to promote the new didactic projects and activities. Nevertheless, more actions are needed, not only at the local level, to raise awareness among institutions and teachers.

Finally, the success of marketing initiatives that led to the creation of three different territorial labels for honey, agri-food products, and tourist reception facilities, points out that the involvement of locally based small and medium-sized enterprises represents a key strategy to be continuously developed and strengthened in order to increase local economic prosperity and rural development. As outlined by Farsani et al. [69] the "*geoproducts not only improve the local economy and present local products, but also educate tourists and popularize geological sciences*". In this context, the internationally recognized "GEOfood brand" [72],

led by Magma UGGp (Norway) since 2013, was created to support the sustainable development of local communities and to foster the cooperation between local farmers and food enterprises within UNESCO Global Geopark territories. Moreover, several geoparks have their own quality labels for food and other geoproducts, e.g., [73,74] and references therein.

From a broader point of view, to evaluate the societal embedding and the involvement of local communities, e.g., [75] in Beigua UGGp, the PhD project "Study of the geosites and biosites of the Beigua UGGp through the application of the concepts of natural capital and ecosystem services" (PhD Course in Sciences and Technologies for the Earth and Environment-University of Genova) is currently in progress.

Further studies are necessary to extend the results obtained from this work, both to the other geosites officially recognized by the Beigua UGGp, and, more in general, to the overall geodiversity of the geopark area. Moreover, detailed geodiversity inventory and maps in a GIS environment, e.g., [76,77] should be developed and continuously implemented because they may play a crucial role in recognizing the most important geoheritage elements and to improve their valorization and conservation.

**Supplementary Materials:** The following supporting information can be downloaded at: https://www.mdpi.com/article/10.3390/land11101667/s1, Table S1: Quantitative assessment of selected geosites (G and GD) from Beigua UGGp according to the criteria of Brilha (2016); Table S2: Weighted score based on the quantitative assessment of Table S1.

**Author Contributions:** Conceptualization and methodology, P.M. and G.C.; investigation and data curation, P.M., G.C., A.B., M.C.C., L.C. and M.F.; writing and original draft preparation, P.M.; review and editing, P.M., G.C., A.B., M.C.C., L.C. and M.F; funding acquisition, M.F. All authors have read and agreed to the published version of the manuscript.

**Funding:** Contributions by P.M. and L.C. were possible thanks to PRIN (Italian Research Projects of Relevant National Interest); Funding number: 2020542ET7. Contributions by A.B. were possible thanks to PRIN; Funding number: 2017RX9XXY. This research received no other external funding.

**Acknowledgments:** We are grateful to the staff of the Beigua UGGp (Daniele Buschiazzo, Claudia Fiori, Ilaria Mangini, and Mirko Moretti) for providing the iconographic material and the data of the activities carried out from 2011 to 2021 within the geopark. The Authors wish to thank the four reviewers and the Academic Editors of this special issue for their insightful and constructive remarks that improved the quality of the manuscript.

**Conflicts of Interest:** The authors declare no conflict of interest.

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
