# Peer review of "Geosite Assessment in the Beigua UNESCO Global Geopark (Liguria, Italy): A Case Study in Linking Geoheritage with Education, Tourism, and Community Involvement"

_land, doi:10.3390/land11101667_

Round 1

Reviewer 1 Report

Orthography :

³ in the authors address  -missing

Correct  Geopgraphic  in the  chapter 2 title.  A careful orthography correction of the entire text is needed.

Unesco  -  UNESCO

geo-tourism – geotourism

 geo-education- geoeducation

Other recommandations:

Ref 53  Refer as : Bonci, M.C.; Piazza, M.; Briguglio, A.; Castello, G.; Caprioglio, C.; Firpo, M. Tropical forest and coral sea of the Beigua Geopark 601 (Liguria, NW Italy).in Grigorescu, Benton & Hairapetian (editors): Paleontological heritage and geoconservation in UNESCO European geoparks, Geoconservation Res., 2021, 4(2), 586-603.

 A short presentation of the ten  geosites and geodiversity sites  presented in fig.4  -recommended  o be includedafter 2. Geographic and geological background

Acknowledgments –? It is unusual that nobody helped , advised or contributed.  

Reviewer 2 Report

I have read this manuscript with significant interest and found various novel and interesting information. Of course, this paper deserves being published in such an impressive journal as “Land”. However, I see various failures, improving which will make this manuscript much better.

a) The abstract should inform better about the main findings.

b) You have to explain why you have chosen the method by Brilha and not some alternative techniques. At least, you have to cite these alternatives.

c) Sorry, but I can’t understand what do the percentages in Table 3 mean. Can you explain it better?

d) Can you write more about marketing? May be you know some examples from the other pkaces?

e) The list of references is detailed, but more sources published after 2020 can be cited.

Reviewer 3 Report

Overall, the paper fits the theme of the Special Issue. However, the paper needs some work to make it suitable as a journal paper. Currently it reads more like a series of statements in a report. Conceptually, there are issues associated with the use of the term’s ‘geology’ and ‘geodiversity’. Much of what is termed ‘geodiversity’ actually conforms with what geologists would term ’geology”. Additional content on linking key geology with educational function and community involvement is suggested.

With regard to lines 61-63

“In the last two decades several methods and criteria for the qualitative and quantitative assessment of geosites and in particular of geomorphosites, have been proposed [10-29] but, to date, a general accepted method has not yet reached… “

This is a blanket statement that the authors should reconsider as to why this is the case within the context of the assessment of sites that links “key geology”, namely geological heritage, or geoheritage foundational to the UNESCO Global Geopark listing. Namely, geosites important for understanding the key geology with educational function and community involvement – the subject of this paper.  See  Brocx and Semeniuk Geoheritage Tool-Kit method [Brocx M & Semeniuk V 2015 Using the Geoheritage Tool-Kit to identify inter-related geological features at various scales for designating geoparks: case studies from Western Australia. In: E Errami, M Brocx & V Semeniuk (eds), From Geoheritage to Geoparks -Case Studies from Africa and Beyond.  Springer, Amsterdam, 245-259], for example, begins with identifying the key geology, or essential geology of an area as a starting place for qualitative assessment of sites.

Also there needs to be clarification that various methods of geosite selection have been developed for geotourism, as distinct from their geoheritage values for conservation and education. Such sites selected for geoheritage values may or may not be suitable for geotourism and community involvement. While there is no generally accepted method of assessment, there are IUCN Guidelines for inventory development and geoconservation. See Crofts et al 2020 Guidelines for geoconservation in protected and conserved areas | IUCN Library System (https://portals.iucn.org/library/node/49132).

With Regard to Lines 327-329

A Geopark offers a wide range of geological and environmental education activities for students and teachers in schools of all levels (Fig. 6). The catalogue is updated every year with new proposals and include short-term and long-term projects.

How does this compare to other Geoparks in Italy, or in the world?

See also Stoffelen, A. Where is the community in geoparks?: A systematic literature review and call for attention to the societal embedding of geoparks. Area 2020, 52, 97–104, doi:10.1111/area.12549. and

Brocx, M. and Semeniuk, V. (2019) The ‘8Gs’—a blueprint for Geoheritage, Geoconservation, Geo-education and Geotourism. Australian Journal of Earth Sciences, 66 (6). pp. 803-821

Line 64 particularly for the numerical assessment [3]. Please explain why numerical assessment is desirable.

I have undertaken some corrections/edits the manuscript with colour-coding and tagged comments on the submitted MS.

The acronym for UNESCO Global Geoparks is UGGp

Reviewer 4 Report

In general, the paper is well designed and structured with a very interesting topic. However, some parts of the paper should be revised based on the following comments:

In the "Materials and methods" section, you state: "For this work, we selected five geosites and five geodiversity sites chosen as representative of both different geological aspects and major educational and tourism activities carried out in the park over the past decade."  - what was the key for the selection? How do you differ between geosites and geodiversity sites? An explanation would be very beneficial.

For the assessment, you have chosen the method proposed by Brilha (2016). Why this method and not any other proposed by other author(s)? Some indications can be found in the Introduction part, however, I would suggest to be more detailed in this part.

On page 9 (lines 250-252), you state: "With the exception of G5, all the sites studied in this work have good or very good interpretative potential (Table 1 and 2) and are particularly suitable for educational activities with students from schools of all levels and universities. Among them, G1, G3, G4, GD6, GD7 and GD10 are located in areas with a high to very high touristic potential." Are there different interpretations for students/visitors of various school levels? (As it is evident that primary school students require totally different approach of interpretation than university students or participants of geological fieldtrip).

Lines 422-425: "The only exception is site G5, despite the considerable scientific value of the eclogite and meta-rodingite outcrops occurring in the area This is mainly caused by the low didactic and interpretative potential for school students and lay people as well as by the geographical location, which lacks scenically valuable elements, particularly in comparison to the other sites considered in this work." Are there any applicable and effective measures to change this state?

Lines 448 - 452: "In order to create synergetic and innovative actions to attract visitors with different interests and to promote public knowledge about geology [63-64], it would be necessary to develop an effective geotourism management plan and a broader outreach strategy involving not only the geopark municipalities but also the regional authorities and business companies." Could you provide some good examples of effective geotourism management plans from other regions/parts of the world/geoparks?
